# Plasma Nitriding of Inner Surface of Slender Tubes using Small Diameter Helicon Plasma

**DOI:** 10.3390/ma16010311

**Published:** 2022-12-29

**Authors:** Chenggang Jin, Yongqi Zhang, Chen Wang, Manxing Liu, Wenbin Ling, Liang He, Yan Yang, Peng E

**Affiliations:** 1Laboratory for Space Environment and Physical Sciences, Harbin Institute of Technology, Harbin 150001, China; 2School of Electrical Engineering and Automation, Harbin Institute of Technology, Harbin 150001, China; 3School of Physics, Harbin Institute of Technology, Harbin 150001, China; 4School of Physical Science and Technology, Soochow University, Suzhou 215006, China; 5No.208 Research Institute of China Ordance Industries, Beijing 100000, China; 6Department of Physics, Yancheng Institute of Technology, Yancheng 224000, China

**Keywords:** small diameter helicon plasma, 316L tube, inner surface plasma nitriding

## Abstract

A steady-state, high-flux N_2_/Ar helicon wave plasma (HWP) with a small diameter (10 mm) was used to nitride the interior of a slender austenitic stainless steel (ASS) 316L tube at a temperature of 450 °C. N_2_ and Ar were fed to a 500 mm long slender tube with 10 mm inner diameter and were ionized inside the tube using a helicon wave in the magnetic field of 2000 G. The microstructure and depth of the nitrided layers, in addition to the morphology and hardness of the nitrided surfaces, were intensively characterized by employing scanning electron microscopy (SEM), optical microscopy (OM), X-ray diffraction (XRD), energy dispersive X-ray spectroscopy (EDS), as well as microhardness tests. The results confirmed that the nitrided layer consisted primarily of the expanded austenite phase γ_N_, and neither CrN nor iron nitride precipitates. An increasing trend in microhardness was observed in inductively coupled plasma (ICP) and HWP modes; however, the increase in HWP nitriding (up to HV 1820 with a thickness of 14 μm) was approximately 1.5 times greater than that achieved through ICP plasma nitriding. This was owing to the higher N^+^ ion density in the HWP mode. Considering the successful control of N_2_ plasma discharge in a slender tube with a small diameter, this study opens up a new avenue for achieving high-yield nitride layers inside slender tubes.

## 1. Introduction

Austenitic stainless steel (ASS) has been extensively used in industrial production owing to its outstanding physical and mechanical properties, specifically good corrosion resistance, plasticity, and finish, in addition to aesthetic surface appearance [1,2,3]. However, due to low hardness and low wear resistance, ASS has limitations that can seriously affect its lifetime and cause economic and technological losses [4,5]. It is generally recognized that nitriding techniques is helpful for ASS, where a supersaturated solid solution of nitrogen with high hardness is formed by diffusing nitrogen from the surface [6,7,8]. Compared with other nitriding techniques, plasma nitriding is well-established, environmentally friendly, and energy-efficient [9,10].

The inner surface plasma nitriding of ASS tubes is a challenge in industrial applications, particularly for tubes with high aspect ratios, such as gun barrels [11] and nuclear fuel cladding [12]. Moreover, it is difficult to utilize an effective tool to treat uniformly the inner surfaces of tubes with a small inner diameter (D) that makes them immune to various changing conditions [13]. Nitrogen implantation/diffusion inside ASS304 tubes with D = 110 mm was achieved after well-adjusted high-voltage applications at temperatures of approximately 350–400 °C [14]. Plasma nitriding with a cathodic cage, which is the hollow cathode effect-based technique, was employed for the inner surface of a pipe with D = 99 mm and tube length (L) = 140 mm [15]. It coated the internal surface of pipes uniformly. Nitrogen plasma implantation inside 200 mm long ASS 304 tubes with D = 40 mm was achieved by Silva et al. by employing a metal cylindrical sieve (MCS) configuration [16]. The above methods are employed in inner surface plasma nitriding of tubes with L/D in the range 1.4–5; however, application of these methods to slender tubes with L ~ 1000 mm and L/D ~ 100 is difficult. It was demonstrated that ions pass through the middle of the tube and arrive at the end of the tube when L ∼ D due to the dependence of L/D on plasma density [17]. This behavior does not occur in longer tubes with L >> D.

A general characteristic of these sources with high density is that the plasma produced from the tube is decoupled. The plasma production occurs from the remote source region and the plasma is diffused into the process inside the tube. Helicon wave plasma (HWP) drivers are typically used to provide high-density plasma columns with high electron density (>10^19^ m^−3^) over its relatively large variations in magnetic field intensities (10–1000 mT) with diameters in the range 20–3 mm [18]. The difference between HWP and inductive coupled plasma (ICP) is recognized as its electron ionization along the magnetic field for a sufficiently long distance (>1000 mm) [19]. In this study, we aim to develop a high-density helicon plasma source with the smallest diameter (10 mm) and a long plasma column (1000 mm), in addition to characterize its plasma that leads to inner surface plasma nitriding slender tubes with D = 10 mm and L = 500 mm. A pulsed DC voltage was applied to the tubes, resulting in a negative bias voltage (–Vs). It can not only play an active role in enhancing the nitriding efficiency on an inner surface, but also avoid an excessively high temperature of the inner surface owing to ion bombardment, as well as suppress the formation of CrN precipitates. The correlation between chemical composition and hardness of the coating was studied. The motivation of this study is to explore a nitriding layer that is hard and wear-resistant inside ASS 316L tubes, thus can be introduced in application domains for harsh environments.

## 2. Experimental Methods

A plasma nitriding processing system using a self-designed helicon plasma source was used to prepare a nitriding layer on the interior surface of a slender ASS 316L tube. The device could be divided into a plasma source with D = 10 mm and a 2000 mm long plasma–material interaction (PWI) chamber with D = 500 mm, according to the function, as shown in Figure 1. The right helicon half-wavelength water-cooled antenna with m = +1 was placed in the source area, and a water-cooled electromagnet was arranged outside the cavity to provide a steady-state magnetic field. The helicon waves were sufficiently excited by a radio-frequency (RF) power source operating at 13.56 MHz. Before nitriding, a slender tube (outer diameter = 16 mm, D = 10 mm, L = 500 mm) was ultrasonically cleaned successively with acetone, alcohol, and deionized water. The tube was baked at 100 ℃ for 12 h under N_2_ atmosphere. Subsequently, the tube was fixed in the processing area, as shown in Figure 1. The PWI chamber was initially evacuated down to a base pressure of 1 × 10^−5^ Pa. Ar^+^ ion bombardment was used to clean the interior surface of the samples for 15 min under the following conditions: constant Ar (99.9999%) flow rate of 50 sccm, working pressure of 1 × 10^−2^ Pa, RF power of 4.5 kW, and magnetic field of 2000 G.

The precursor N_2_ was fed to the reaction zone through the flow controller and was dissociated under the action of the Ar helicon wave plasma (wave coupling), resulting in the formation of active nitrogen particles with high concentration onto the substrate. Active nitrogen particles diffuse into the substrate under the action of the negative pulsed bias. They chemically react with Fe, forming a nitrided layer on the substrate surface before nitriding. The flow rate of Ar as the helicon wave discharge gas was maintained at 50 sccm; the flow rate of N_2_ (99.9999%) as the reaction gas was maintained at 120 sccm; the working pressure of the nitriding treatment was maintained at 0.3 Pa; the magnetic field strength was maintained at 2000 Gs; and the RF power was varied over the range of 1 to 4.5 kW. The slender tube had a pulsed bias voltage of –400 V with a frequency of 10 kHz, duty ratio of 20%, and tube temperature of 450 °C, which was measured using a thermocouple in real time. The nitriding experiment was maintained for 3 h. After finishing the nitriding experiments, the tubes were cooled using N_2_ to approximately 25 °C. The specimen (15 mm length) was cut off from the end of the outlet in the tube before and after plasma nitriding via wire electrical discharge. Subsequently, the specimens were then sectioned with a width of 1 mm, and the sections were used to measure the surface properties.

Optical emission spectroscopy (OES) was conducted using a grating spectrometer (Princeton Instruments, Trenton, NJ, USA, HRS-750S) and an intensified charge-coupled device (ICCD) camera (Princeton Instruments, Trenton, NJ, USA, PM4-1024i) equipped with a 74-UV silica-collimating lens. The OES spectra were acquired between 200 and 900 nm in wavelength range, with a step size of 0.03 nm. A mass and energy analyzer, electrostatic quadrupole plasma (EQP) (Hiden Analytical Limited, Warrington, UK) was employed to monitor the ion energy distribution (IED) and the particle mass distribution near the inner surface of the tube. The probe was placed into a 6 mm diameter drilled hole on the substrate in the PWI region chamber.

The morphological properties and depth of the nitrided layers were characterized by employing optical microscopy (OM, FS70L, Mitutoyo, Kawasaki, Japan) and field emission scanning electron microscopy (FE-SEM, SU8010, Hitachi, Tokyo, Japan). Chemical composition of the nitrided layer was analyzed by employing energy dispersive X-ray spectroscopy (EDS, Bruker XFlash 6130, Karlsruhe, Germany). Elemental composition and chemical bonding were investigated by employing x-ray photoelectron spectroscopy (XPS, ESCALAB 250XI, Waltham, MA, USA). High-resolution XPS recorded spectra were acquired in ultra-high vacuum utilizing a monochromatic Al Kα radiation (hυ = 1486.6 eV) as the X-ray source, in addition to the equipped in situ Ar^+^ ion etching. During measurement, the spectra were acquired at a pressure of 6.6 × 10^−10^ mbar. In addition to the selected scan parameters, the instrumental energy resolution of the XPS instrument was 0.43 eV, defined as the full-width half maximum (FWHM) of the Ag 3d5/2 peak. Complementary work function measurements were conducted via UPS using the He I (21.2 eV) line without a monochromator. The analyzed sample area was a circle with a diameter of 650 μm. The electron emission angle was 58°. The sample sputter-etched period was 10 min, the energy of Ar^+^ was 1000 eV, and the incidence angle was 40 degrees from vertical. The charge neutralized was used during the XPS experiment. The charge referencing method used was the work function method [20]. Through the experiment of UPS, the work function of the samples was obtained and the value was φsa = 4.55 eV. Thus, the C 1s referencing was 285.03 eV. In order to analyze the microstructure of the nitrided layers inside the tube, X-ray diffraction was performed using a Bruker D8-Advance diffractometer. The recorded XRD pattern was acquired using the equipped Cu-Kα radiation (λ = 0.15418 nm) in the typical θ–2θ pattern. The microhardness and hardness profile of the nitrided layer were determined using a dynamic ultra-micro hardness tester (DUH-211, Tokyo, Japan) in load−unload mode. The indentation depth was 10 μm and the load speed was 2.22 mN/s. The indenter type was a triangular pyramid indenter with a tip angle of 115°. The indentation on each sample was tested 12 times in different areas to ensure statistical reliability.

## 3. Results and Discussion

### 3.1. Phase and Microstructure Analysis

Figure 2 presents the typical surface and cross-sectional OM of the original sample and the samples nitrided in ICP mode with an RF power of 1 kW and HWP mode with an RF power of 4.5 kW, marked as S1, S2, and S3, for 3 h. The thicknesses of the nitrided layers are determined according to the cross-sectional OM images of the layers. With an increase in the RF power from 1 kW (ICP mode) to 4.5 kW (HWP mode), the thickness of the nitrided layer increases from 7 μm to 14 μm. The distinct variation in the surface morphology before and after nitriding treatment is the evidence for ion bombardment. In addition, the average diameter of these particles is significantly smaller than that of the particles shown in Figure 2a.

Figure 3a,b show the surface SEM of Samples S1 and S3, respectively. After HWP nitriding, the surface of Sample S3 becomes smooth compared to Sample S1, which is consistent with the results of the OM images. From the energy dispersive X-ray spectroscopy (EDS) elemental mapping result shown in Figure 3c, Fe and N are well-distributed in the resultant nitrided layer, evidencing that a high concentration of N of 17 at. % (atomic percentage) was uniformly doped into the inner surface of the slender tube for Sample S3. The cross-sectional morphology as well as N concentration depth profile of the nitrided layer of Sample S3 are presented in Figure 3d. It is worth noting that the shape of the obtained N distribution profile near the surface in this work exhibits a plateau type, showing a good consistency with the proposed “trapping–detrapping (TD)” model [21]. With respect to the TD model for the nitrogen diffusion mechanism in plasma-nitrided ASS, a highly N-enriched layer is created with an N concentration that is equal to or even greater than the Cr concentration when mobile N atoms are trapped by the Cr atoms forming a chemical bond.

The XRD results shown in Figure 4 confirm that Sample S1 has a single austenitic phase (γ), which exists as a face center cubic (fcc) structure with variation at 43.37° and 50.57°. As depicted in Figure 4, the significant difference can be clearly seen between detected diffraction peaks from the nitrided Samples (S2 and S3) and the un-nitrided (S1, original austenitic stainless steel). Specifically, a characteristic set of broader peaks that appear at a lower 2θ angle side than those for the un-nitrided Sample S1 are presented in the patterns of the nitrided Samples. They correspond to γN, named “a high nitrogen fcc phase”, which is commonly known as a solid solution supersaturated with nitrogen that has a disordered fcc structure [22,23]. The γ_N_ phase is frequently referred to as an “expanded austenite” since it has a greater lattice parameter compared to austenite [24]. As a result, the dominant phase in the nitrided layer is the γ_N_ phase. Moreover, the diffraction peaks increase in intensity with an increase in the RF power, which is consistent with the large N atom concentration observed in the EDS results. And it is worth noting that no evidence of CrN or α phases can be seen adjacent to the principal peak (γ_111_).

During the plasma nitriding process of 316L, Cr-N and Fe-N clusters can be produced nano-scale in the heterogeneous expanded austenite phase. This is due to the overlapping and broadening of diffraction peaks, particularly the nanostructured secondary phases; it is rather difficult to determine the presence of more than one precipitate type, and a high-resolution measurement technique is, hence, required. The characterization of 316L stainless steel inner surface layers after plasma nitriding treatment was investigated by employing XPS. Before XPS measurements, the surface of all samples was etched by ~100 nm using an Ar^+^ beam to eliminate the residual contaminants. The chemical composition (in atomic concentrations) of the inner surface is calculated in Table 1. After Ar ion etching, the N content increases from 16.5% to 20.6%, and O content decreases from 2.9% to 2.1% with ICP and HWP nitriding, respectively. Therefore, we can conclude that it is sufficiently efficient to eliminate the top surface contamination by scraping the top with 100 nm off the surface layer. The survey spectra of XPS measurement of sample S3 is presented in Figure 5a. The detected carbon and some of the oxygen species come from the adventitious contamination whilst the oxides and hydroxides come from the reaction layer in 316L after exposure to atmosphere environment. Figure 5b shows the carbon peak C 1s of 285.03 eV indicating the presence of sp^2^ carbon. The oxygen peak O 1s appears to consist of two species corresponding to presence of metal oxides (530 eV) and organic C-O (531 eV), as shown in Figure 5c. Figure 5d shows the XPS spectra of N 1s for sample S3 after nitriding treatment. The corer level spectra display mainly two peak features at 396.6 eV and 397.5 eV, corresponding to the reported binding energies of the N 1s peaks of CrN and Cr_2_N, respectively [25]. The N 1s peak at 399.8 eV is thought to be connected to either adsorbed N_2_ [26] or oxidized nitrogen [27] but not the formation of chromium nitride. XPS results confirm the formation of Fe-N and Cr-N compounds in sample S3, which is not determined in the XRD results. Figure 5e shows the XPS spectra of Cr 2p_3/2_ for sample S3 after nitriding treatment. The corer level spectra display mainly three peak features at 574.0 eV, 574.8 eV, and 576.4 eV, corresponding to the reported binding energies of Cr, Cr-N, and Cr-O, respectively [25]. Figure 5e shows the XPS spectra of Fe 2p_3/2_ for sample S3 after nitriding treatment. The corer level spectra display mainly two peak features at 706.7 eV and 707.4 eV, corresponding to the reported binding energies of Fe and stoichiometric iron nitrides, respectively [7]. Figure 5g shows the carbon peak Ni 2p of 586.6 eV indicating the presence of metal Ni. The 316L samples after nitriding treatment are dominated by oxides owing to their high chemical stability. With Ar-ion bombardment, the formed nitrides deeply penetrate into the surface and coexist with the surface oxide film, resulting in the N-enriched oxide coating. The passive coating on the surface of 316L consists mainly of Cr-O and C-O.

### 3.2. Hardness Measurements

The surface and subsurface microhardness of the tested specimen (at the cross-section) can be assessed by microhardness measurement using a Vickers diamond indenter with approximately 40 μm intervals. Figure 6 illustrates the cross-sectional hardness of Samples S2 and S3, which shows that the samples were significantly strengthened by plasma nitriding compared to Sample S1, that is, they remains at 115 % of the tube hardness (~HV200) at a depth of 30 μm. It is clear that the load on the surface of the S3 nitrided layers exhibits the maximum value (~HV1820), with the same indentation depth as that on Sample S2 nitrided layers (~HV1208), which was approximately 1.5 times more. This suggests high levels of hardness in Samples S2 and S3 nitrided layers. The improved surface hardness can be ascribed to the diffusion of nitrogen atoms in the stainless steel matrix and elimination of the sputtering phenomena from the samples, forming an extremely high hardness phase (γN) with an ordered fcc structure. This is in agreement with the XRD results.

### 3.3. Plasma Diagnostics

To understand why higher concentrations of nitrogen are incorporated into the surface of ASS 316L slender tubes in HWP mode, the N_2_/Ar plasma was further analyzed through OES, as illustrated in Figure 7. Figure 7a shows the emission spectrum recorded in the wavelength range of 350–450 nm. The intensities of N_2_, $N_{2}^{+}$, and Ar^+^ in the HWP mode are very high compared to those in ICP mode, evidencing that they are the major component of the plasma species. Figure 7b shows emission lines in the range 730–750 nm. In HWP mode, the emission lines of N at wavelengths of 742.4 nm, 744.2 nm, and 746.8 nm are presented due to the high plasma density. Due to the strong N–N bonding, which leads to the dissociation of N_2_, there exists a nonlinear relationship between the electron density and NI intensity that gives rise to the N_2_ plasma discharge. A possible elucidation is that, in spite of the constant electron temperature, N_2_ is virtually entirely dissociated or the dissociation rates adopt a plateau HWP mode. Thus, NI suitably describes the behavior of atomic N-based plasma. However, in the experiment, the Fe line (371.99 nm) could not be found. According to the OES findings, neutral Fe could not be a necessary species in plasma nitriding treatment.

The typical ion mass spectra are measured using the EQP detector during plasma discharge in different modes. The EQP results indicate that the main ion species are $N_{2}^{+}$, Ar^+^, and N^+^. Figure 8 shows the ion energy distributions (IEDFs) of $N_{2}^{+}$, Ar^+^, and N^+^ ions in the ICP and HWP modes; the shift of ion energy and evolution of peak profiles can be obtained. As shown in Figure 8b, the IEDFs of Ar^+^ ions show the presence of a low energy peak, as well as a high energy peak, corresponding to the bulk ions and ion beam, respectively. This is a typical helicon mode compared to ICP with a single peak. The ion densities of $N_{2}^{+}$, Ar^+^, and N^+^ in the HWP mode are much higher than those in the ICP mode, which is confirmed by the OES results. The N^+^/$N_{2}^{+}$ increases rapidly from 0.12 to 2.25 as the discharge mode changes from ICP to HWP, which should be owing to the interaction between particles of different elements.

According to the results, the influence of N_2_/Ar on the incorporation of N into the inner surface of the slender ASS 316L tube can be concluded. N_2_/Ar HWP treatment as a technique to nitride the inner surface of the ASS 316L tube confirms that N^+^ is an efficient technology for treatment of material surface. N^+^ plays a significant role in chemical reactions owing to its high reactivity; thus, N^+^ concentration is significant in the nitrided layer.

## 4. Conclusions

In conclusion, the inner surface of a 500 mm long slender tube with an inner diameter of 10 mm was nitrided at a temperature of 450 °C using an N_2_/Ar HWP process. The modification of the metal tube inner surface with such a large aspect ratio (50:1) was not achieved in other attempts. As a result, all the inner surfaces of tubes with high aspect ratio were nitrided. A large amount of N was incorporated into the inner surface of the tube with 17% N concentration according to the EDS results. For the nitrided HWP, a nitride layer with a thickness of 14 μm was produced on the surface, approximately two times higher than in the ICP nitrided layer. Surface hardening resulting from nitriding can enhance the microhardness up to HV1820, which is nearly nine times higher than that of the untreated surface. It can be concluded that in this study, the enhancement in microhardness is primarily attributed to the creation of γ_N_ phase with compressive residual stresses and microdistortions in the high N-containing layer as a result of high concentration of N^+^. This solution can be applied to improve the hardness of the inner surface of slender tubes, which is particularly helpful for future industrial applications, such as nuclear fuel cladding.

## Figures and Tables

**Figure 1 materials-16-00311-f001:**
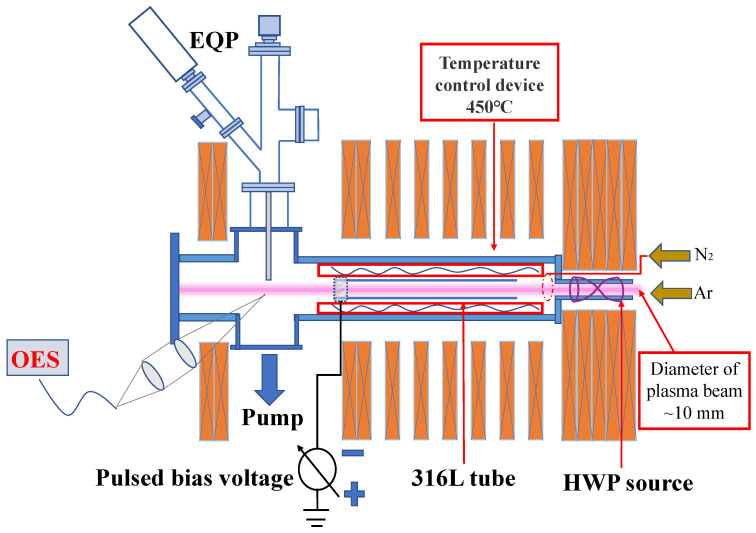
Schematic of the helicon plasma system.

**Figure 2 materials-16-00311-f002:**
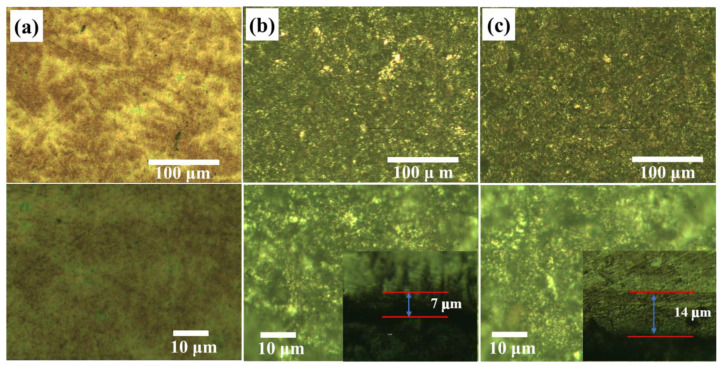
OM surface morphologies and cross-sectional microstructures of the samples (**a**) S1, (**b**) S2, and (**c**) S3.

**Figure 3 materials-16-00311-f003:**
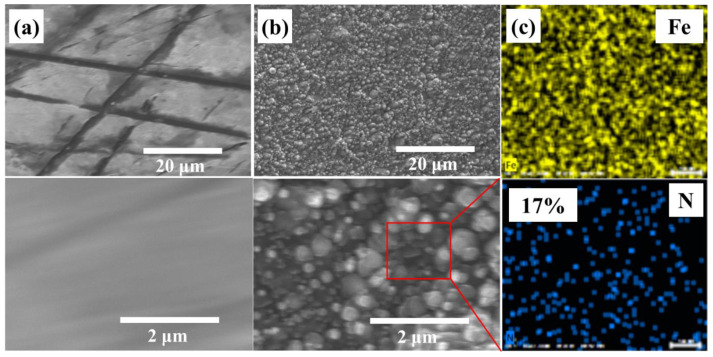
SEM micrographs of the inner surface of (**a**) Sample S1 and (**b**) Sample S3; (**c**) EDS of Sample S3; (**d**) cross-section SEM micrograph obtained perpendicularly to the N diffusion direction from Sample S3 with 14 um nitrided layer.

**Figure 4 materials-16-00311-f004:**
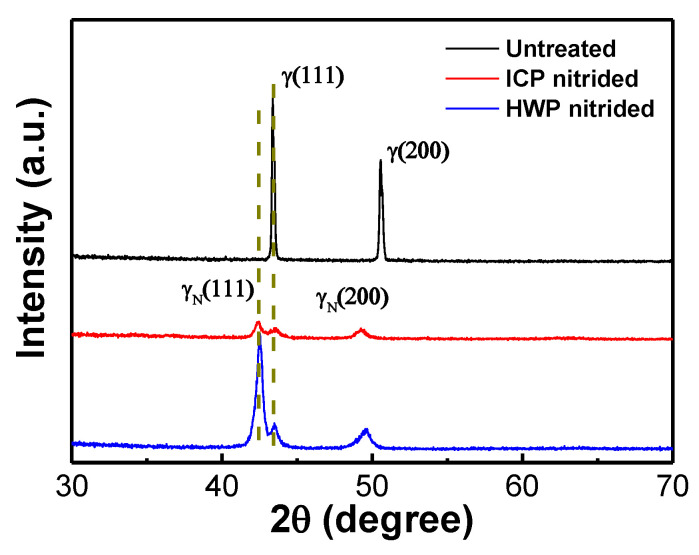
XRD patterns of Samples S1–S3.

**Figure 5 materials-16-00311-f005:**
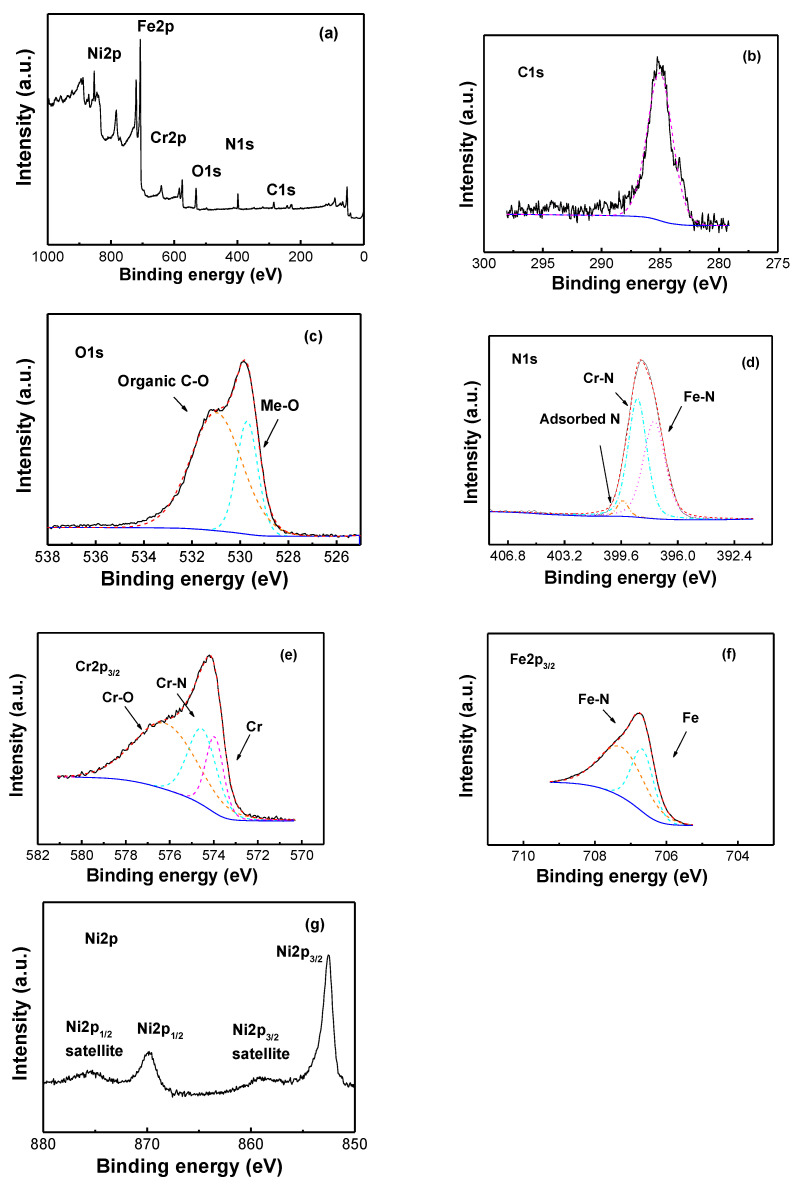
X-ray photoelectron spectroscopy spectra of Sample S3: (**a**) survey spectra, (**b**) C1s, (**c**) O1s, (**d**) N1s, (**e**) Cr2p_3/2_, (**f**) Fe2p_3/2_, (**g**) Ni2p.

**Figure 6 materials-16-00311-f006:**
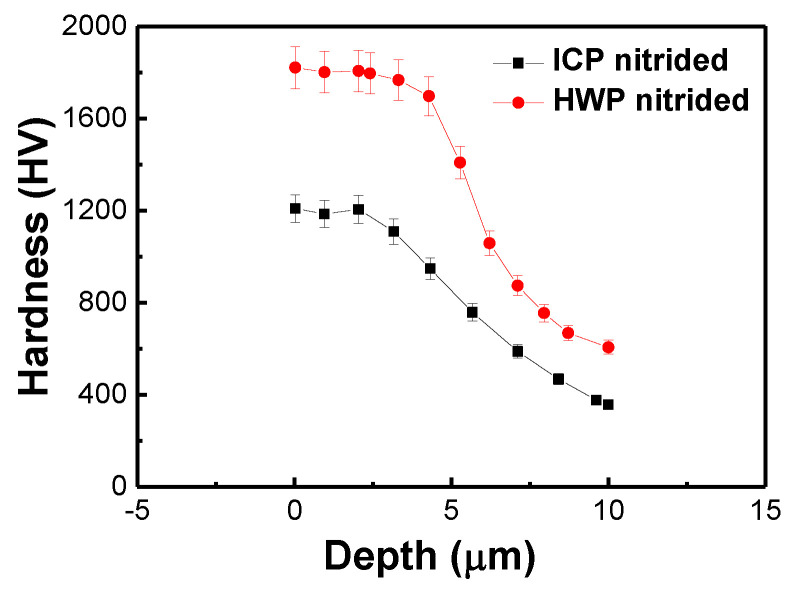
Microhardness profiles of Samples S2 and S3.

**Figure 7 materials-16-00311-f007:**
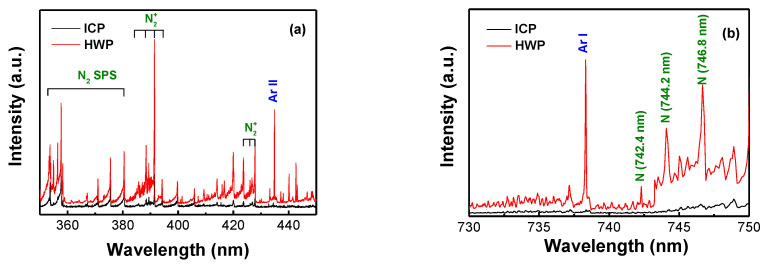
OES spectrum observed during plasma nitriding process in: (**a**) wavelength range 350–450 nm, (**b**) wavelength range 730–750 nm.

**Figure 8 materials-16-00311-f008:**
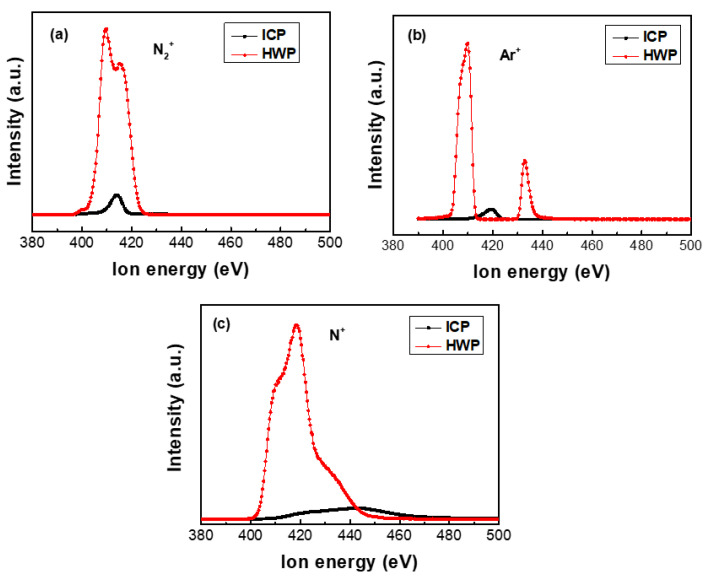
Normalized ion energy distributions: (**a**) $N_{2}^{+}$, (**b**) Ar^+^, and (**c**) N^+^ ions.

**Table 1 materials-16-00311-t001:** Inner surface composition of ICP nitride and HWP nitride samples (in at.%).

Sample	C	O	N	Fe	Cr	Ni
ICP nitride	2.7	2.9	16.5	52.9	15.2	9.8
HWP nitride	2.2	2.1	20.6	50.8	14.6	9.7

## Data Availability

Data will be made available on request.

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
