# Peer review of "Plasma Nitriding of Inner Surface of Slender Tubes using Small Diameter Helicon Plasma"

_materials, 2022, doi:10.3390/ma16010311_

Round 1

Reviewer 1 Report

The authors of this paper have conducted an interesting investigation into the possibilities of nitriding the inner surfaces of deep holes in austenitic steel. The paper is clearly divided into main sections, and the sections properly describe the experiments that the authors performed on the experimental samples. The results of the experiments are properly described and justified. 

Comments:

For Figure 2, photos of the surface are shown but the cross-sectional images are unclear and confusing. It would be better to give pictures of the layer thickness separately as in Figure 3d.

Hardness measurements : the authors describe that they measured surface and subsurface hardness but do not specify the measurement conditions. Furthermore, I would like to explain how it is possible to measure the surface hardness in cross-section while keeping the basic condition of placing the indentation at least 2 diagonal lines of the indentation.

Further to the evaluation methodology it is not clear to me how the authors measured the properties along the length of the hole, from the text they refer to one sample but at the very least it would be useful to show whether the nitriding is the same along the length of the hole and whether there are differences in thickness at the edge and in the centre of the sample.

Reviewer 2 Report

The manuscript is about an interestic topic and the authors provide evidence of forming a nitriding layer in the inner surface of a tube with certain characteristics. They extensively characterized the layer through different techniques. However, some improvements must be made:

1. The treatment temperature is 450 °C, but the authors can provide a little more explanation of why this temperature was choosen? Why not higher or lower?

2. Cross section micrographies of the layers of Figure 2b and 2c must be improved. Is not quite clear what is the nitride layer in the bottom inset of Fig. 2b The layers were etched for thickness measurements?

3. The micrography of Figure 3d looks that was processed/filtered? Please specify. If not, in what SEM mode was acquired?

4. In the hardness profiles of Fig. 6 seems that the first indentation is at 0 µm from the surface, which would be an incorrect procedure.

https://www.emcotest.com/en/the-world-of-hardness-testing/hardness-know-how/applications-tips/general-tips/minimum-distance-between-test-points-and-to-the-specimen-edge/

5. Please provide an optical micrograph along the layer were indentations can be seen. Additionally, please include error bars in each point, in the current plot it seems like a single indentation was performed at each distance.
